# Mechanisms and Drivers for the Establishment of Life Cycle Complexity in Myxozoan Parasites

**DOI:** 10.3390/biology9010010

**Published:** 2020-01-01

**Authors:** Martina Lisnerová, Ivan Fiala, Delfina Cantatore, Manuel Irigoitia, Juan Timi, Hana Pecková, Pavla Bartošová-Sojková, Christian M. Sandoval, Carl Luer, Jack Morris, Astrid S. Holzer

**Affiliations:** 1Institute of Parasitology, Biology Centre of the Czech Academy of Sciences, Branišovská 31, 37005 České Budějovice, Czech Republic; jedlickovamarti@seznam.cz (M.L.); fiala@paru.cas.cz (I.F.); pecka@paru.cas.cz (H.P.); sojkova@paru.cas.cz (P.B.-S.); 343gs@email.arizona.edu (C.M.S.); 2Department of Parasitology, Faculty of Sciences, University of South Bohemia, Branisovska 31, 37005 Ceske Budejovice, Czech Republic; 3Laboratorio de Ictioparasitología, Instituto de Investigaciones Marinas y Costeras (IIMyC), FCEYN, UNMdP-CONICET, 7600 Mar del Plata, Argentina; cantator@mdp.edu.ar (D.C.); mmirigoitia@mdp.edu.ar (M.I.); jtimi@mdp.edu.ar (J.T.); 4Undergraduate Biology Research Program (UBRC), University of Arizona, 1200 E. University Blvd., Tucson, AZ 85721, USA; 5Mote Marine Laboratory, 1600 Ken Thompson Parkway, Sarasota, FL 34236, USA; caluer@mote.org (C.L.); morrisj@mote.org (J.M.)

**Keywords:** Chondrichthyes, myxozoa, cnidaria, co-phylogeny, co-diversification, phylogeography, migration, feed-integration

## Abstract

It is assumed that complex life cycles in cnidarian parasites belonging to the Myxozoa result from incorporation of vertebrates into simple life cycles exploiting aquatic invertebrates. However, nothing is known about the driving forces and implementation of this event, though it fostered massive diversification. We performed a comprehensive search for myxozoans in evolutionary ancient fishes (Chondrichthyes), and more than doubled existing 18S rDNA sequence data, discovering seven independent phylogenetic lineages. We performed cophylogenetic and character mapping methods in the largest monophyletic dataset and demonstrate that host and parasite phylogenies are strongly correlated, and that tectonic changes may explain phylogeographic clustering in recent skates and softnose skates, in the Atlantic. The most basal lineages of myxozoans inhabit the bile of chondrichthyans, an immunologically privileged site and protective niche, easily accessible from the gut via the bile duct. We hypothesize that feed-integration is a likely mechanism of host acquisition, an idea supported by feeding habits of chimaeras and ancient sharks and by multiple entries of different parasite lineages from invertebrates into the new host group. We provide exciting first insights into the early evolutionary history of ancient metazoan parasites in a host group that embodies more evolutionary distinctiveness than most other vertebrates.

## 1. Introduction

Parasite life cycle changes generally have adaptive components and are not purely accidental [1]. Changes in life cycle complexity, i.e., the acquisition of a new host group can either be a necessary response to changes in external environmental conditions or the outcome of the differential success of alternative transmission strategies under stable external conditions, and is usually associated with evolutionary advantages [2].

Myxozoans are a group of cnidarians that are estimated to have emerged approximately 601–700 million years ago (mya), and there is compelling evidence from phylogeny and life history, that they first settled as parasites of aquatic invertebrate hosts [3]. Following the occurrence of aquatic vertebrates on Earth, myxozoans implemented fish, which emerged 410–447 mya [4,5], as secondary hosts in their life cycles. This event fostered massive host-associated biodiversification [3] and likely presents the main reason for the distinct success of the Myxozoa, when compared with other parasitic cnidarians. Myxozoans are extremely reduced in size, and only their polar capsules, homologues of stinging cells of free-living cnidarians, give away their affiliation with the phylum and support their inclusion into the Cnidaria (e.g., [6,7,8]). Though predominantly known as pathogens of fishes, myxozoans can infect a wide variety of vertebrate groups, including amphibians, reptiles, birds, and even terrestrial mammals ([9] and references therein), and they appear to be extraordinarily flexible with regard to host switches from fish to completely unrelated groups such as mollusks or platyhelminths [10,11,12,13]. Not surprisingly, it appears that fish were acquired as secondary hosts multiple times during myxozoan evolution [3,14].

Myxozoans are highly diverse, representing about one fifth of the cnidarian biodiversity known to date [15]. Since their diversification is strongly linked with vertebrate host speciation [3], the evolutionary event of secondary host acquisition is of particular importance, yet it is unknown which vertebrate hosts were settled first and by what mechanism the successful life cycle expansion event was implemented in this group of parasites, as well as why complex life cycles did not evolve in other lineages of parasitic cnidarians. While agnaths are poorly explored as hosts for myxozoans (single record from hagfish [16], no DNA sequences), the oldest vertebrate group known to harbor an important number of myxozoans is cartilaginous fish, i.e., sharks, rays, skates, and chimaeras e.g., [14,16,17,18,19,20]. To date, 46 species belonging to seven genera have been formally described from Chondrichthyes (Appendix A), with phylogenetic positions of 19 taxa determined, using 18S rDNA sequences. The most basal branches in two out of the four major clades of myxozoans are represented by species from cartilaginous fishes [3], while the remaining clades are devoid of lineages from Chondrichthyes, potentially due to missing data. 

Worldwide, over 1250 species of Chondrichthyes are known [21], and they are by far the most evolutionarily distinct radiation of all jawed vertebrates, with the average species embodying 26 million years of unique evolutionary history [22]. With the exception of the living fossil lineages Coelacanthiformes and Dipnoi, only Agnatha embody more evolutionary history per species than the average chondrichthyan [22]. Hence, this class of ancient vertebrates is of special importance to reconstruct the early evolutionary history of myxozoans in vertebrates. However, Chondrichthyes are among the most imperiled marine organisms, with up to a quarter facing increased risk of extinction [23], together with the potential for their evolutionarily distinct parasite fauna. 

Considering the paucity of myxozoan sequence data from cartilaginous fishes, and the great potential of such data to unveil ancient evolutionary events that shaped this parasite group, we screened sharks and rays from seven different orders for myxozoan infections. We more than doubled existing 18S rDNA sequences of myxozoans from these hosts, and used phylogenetic, cophylogenetic, and character mapping methods to unveil the early history of myxozoan settlement in evolutionarily ancient fishes. We aimed at identifying phylogenetic origins and diversity hotspots with regard to geography as well as host groups and in relation to host migratory behavior. Finally, we discuss the potential mechanisms for establishing a complex life cycle in predominantly large, predatory fishes.

## 2. Materials and Methods

### 2.1. Isolation and Characterization of Myxozoans from Chondrichthyes

Twenty one species of sharks and rays, belonging to 7 different orders (Appendix A) were obtained from 3 geographical areas: the Gulf of Mexico (coast of Florida, USA), the Atlantic off South Carolina (USA), and the Atlantic off Mar del Plata (Argentina). Gulf of Mexico specimens *Rostroraja eglanteria*, *Sphyrna tiburo*, and *Carcharhinus limbatus* were collected using trotline or gill nets in local waters off Sarasota, FL, under Special Activities License (SAL) research collecting permits issued by the Florida Fish and Wildlife Conservation Commission. All other specimens were commercial catches landed in Charlston Harbor (*Rhizoprionodon terraenovae*) and at the port of Mar del Plata (all other species), after transport on ice by the fishing vessels, and were dissected at the fish processing plant. 

Fish tissues were analyzed fresh or post freezing (storage). Microscopic preparations of bile/liver and kidney of all species as well as muscle of some specimens were screened microscopically at 400× magnification and the same tissues/liquids were stored in TNES urea for DNA extraction using a conventional phenol-chloroform protocol [24]. Depending on access and time, samples from several other organs (gills, skin, heart, stomach, intestine, gonads) were also screened visually and by PCR, in 31% of fishes. Digital images of unfixed myxozoan spores were obtained whenever visible and their morphology was compared with published records to determine unique morphological characteristics. As 18S rDNA is the most taxa-rich sequence collection of myxozoan DNA sequences presently available, extracted DNA was subjected to PCR amplification of myxozoan 18S rDNA, using an optimized methodology based on a combination of universal and more specific primers [24,25]. PCR products were separated by electrophoresis on 1% agarose gels. Bands were cut, purified (Gel/PCR DNA Fragments Extraction Kit, Geneaid Biotech Ltd., New Taipei City, Taiwan) and commercially Sanger sequenced (www.seqme.eu). 

### 2.2. Novel Lineages, Their Phylogeny, and Diversity

To determine phylogenetic relationships between new unique 18S rDNA sequences and published ones, comprehensive alignments were prepared that included all available sequences of myxozoan isolates from cartilaginous hosts (21 new and 18 published taxa) as well as selected myxozoan representatives from all previously defined large clades (total alignment of 137 taxa, 1746 bp; Appendix A). Nucleotide sequences were aligned applying Mafft version 7.017 [26] implemented in Geneious Prime v 11.1 [27], using the E-INS-i algorithm, with a gap opening penalty of 2.0. Alignments were edited to remove highly variable sections. Maximum likelihood (ML) analyses were performed using RAxML v7.2.8 [28] with the GTR + Γ model of nucleotide substitution. Maximum parsimony (MP) analyses were done in PAUP* v4.0b10 [29], using heuristic search with random taxa addition, Ts:Tv = 1:2 and random addition of taxa. Bootstrap support was calculated from 1000 replicas. 

In order to determine whether a correlation exists between parasite diversity and host migratory behavior, we quantified the number of parasite and host species in known infected chondrichthyan orders and defined the migration status of each species into categories (Appendix A). Movement patterns are frequently categorized as ‘migratory, nomadic, or resident’, based on definitions in Mueller and Fagan [30], however, by definition this approach would have left us with a too small dataset (n = 3) for statistical analyses in the ‘nomadic’ category. Furthermore, some species such as *Squalius acanthias* can cover either extremely large distances or migrate smaller distances seasonally while others even reside as local population. We hence created 3 categories based on ranges of distances travelled per year (resident/movements up to 500 m radius (1), >500 m but <1000 km movements (2), and >1000 km migrations), taking into account that some migratory species can travel the same distance as nomads, potentially encountering the same number of parasites. Migratory behavior was categorized based on the information available in the IUCN Red List of Threatened Species https://www.iucnredlist.org or by using data from peer-reviewed studies using transponders and tracking the movements of individuals. Subsequently, we analyzed diversity and migration data using generalized linear models [31] with a Poisson distribution (logarithmic link function) and tested the significance of correlation by the χ^2^ criterion. All analyses were performed in R [32]. 

### 2.3. Host-Parasite Cophylogeny and Geographic Character Correlation in Chloromyxum spp.

The largest number of parasite 18S rDNA sequences obtained from a single lineage (23 taxa) and a diverse spectrum of Chondrichthyes (9 orders) is represented by *Chloromyxum* spp. This lineage likely represents a single entry into secondary hosts followed by radiation predominantly in cartilaginous fishes (see Section 3.2). We analyzed this clade in more detail with regard to potential host-related and geographic radiation. For cophylogenetic studies, parasite trees of *Chloromyxum* spp. were produced as stated above, with all nucleotide positions included in phylogenetic analyses (1901 bp). To obtain a reliable host phylogeny, a DNA supermatrix of 15 coding and non-coding regions (60,121 bp; [22]) was used for the related 21 host species, according to availability of these genes on GenBank (March 2019). Phylogeny estimates were performed as described above. To compare host and parasite phylogenies, we used an event-based tree reconciliation method, CoRe-PA v 0.5 [33], with data-based cost estimation. Reliable trees are a prerequisite for meaningful tree-based cophylogenetic analyses. We used the best ML host and parasite trees (RAxML), acknowledging low support for several branches in the *Chloromyxum* spp. tree (Appendix A) and as a consequence also analyzed partially unresolved trees (Appendix A). To further support codivergence of host and parasite lineages, we applied a global fit estimate of cophylogeny based on a matrix of raw patristic distances and hence a method that is independent from tree topologies, using PARAFIT [34], implemented in the APE package, version 3.4 in R. We additionally performed historical biogeographical reconstructions in order to illuminate the evolutionary history of chloromyxids in sharks and rays, in space and time. We used dispersal-vicariance analyses (DIVA) [35,36] and tested for geographic-phylogenetic character correlations using S-DIVA [37], thereby reconstructing ancestral states.

## 3. Results

### 3.1. Preferred Cartilaginous Hosts of Myxozoans

All major chondrichthyan lineages, the subclass Holocephali (chimaeras) and all three superoders within the subclass Elasmobranchii, i.e., Batoidea (rays, skates, and sawfish), Galeomorphii (modern sharks) and Squalomorphii (ancient sharks) were found to harbor myxozoan parasites (Figure 1), with 70 taxa known to date (Appendix A) and at least seven independent historic entries into cartilaginous hosts (Figure 2), based on limited molecular data available to date (41 distinct 18S rDNA sequences). We discovered three new lineages of myxozoans in elasmobranchs (*Sphaerospora*, *Ortholinea*, and *Parvicapsula*), amounting to a total of 6 lineages in this group (Figure 2). *Chloromyxum* spp. occur in 14 families of elasmobranchs (Figure 1) and represent the most diversified lineage of myxozoans in these hosts, accounting for 38 species, exclusively found in elasmobranchs. The second most diverse genus is *Ceratomyxa*, with 20 species from sharks, rays, and skates, which cluster in a mixed clade with *Ceratomyxa* spp. from teleosts. In contrast to elasmobranchs, in chimaeras, only two species are known, *Bipteria vetusta* which occupies a unique basal lineage (Figure 2) and *Ceratomyxa fisheri* [16] (no DNA sequence data available). Kidneys and gall bladders/liver were commonly found infected with myxozoans, some muscle samples also harbored parasites, however, no other organs tested positive by microscopy or PCR-based DNA analyses.

The family *Carcharhiniformes* (ground sharks) as one of the most diverse families of elasmobranchs (280 species; [22]) presently exhibits the highest diversity and number of myxozoans (7 genera, 32 species; Appendix A) and we observed a clear correlation between the number of myxozoans described and the host diversity in a given order of Chondrichthyes (χ^2^ = 21.855, df = 12, *p* < 0.001; Appendix A). Larger numbers of parasite species were observed in migratory host species than in species with a low geographic range or non-migratory behavior (Appendix A), however, the correlation was not significant (χ^2^ = 4.36, df = 2, *p* = 0.11). 

### 3.2. Origin of Lineages and Comparison with Phylogenetic Position of Teleost Congeners

The addition of 22 new and unique 18S rDNA sequences to the existing 19 led to a change in the arrangement of the major myxozoan clades (Figure 2), due to the addition of distinct basal lineages. The phylogenetic tree of myxzoans is characterized by four well-supported main lineages, the bryozoan-infecting malacosporeans, the *Sphaerospora sensu stricto* clade as well as a polychaete- and and oligochaete-infecting clade of myxozoans (recently reviewed in [3]). Here, myxozoans belonging to the *Sphaerospora sensu stricto* clade were sequenced for the first time from elasmobranchs. This clade is normally consistently positioned basal to oligochaete- and polychaete-infecting lineages [3,38,39] while it was reconciliated as sister to the polychaete-infecting clade, with 82% bootstrap support in both, maximum likelihood and maximum parsimony analyses, in the present study. 

*Bipteria vetusta* from *Chimaera monstrosa* represents the most basal lineage in polychaete-infecting myxozoans, similar to *Chloromyxum* spp. from elasmobranchs, which form the most basal branch of all oligochaete-infecting myxozoans. *B. vetusta* and all *Chloromyxum* spp. are parasites of the biliary system of their cartilaginous hosts and hence share a common host tissue location, despite being phylogenetically independent lineages. At present, members of the *Bipteria* and the *Chloromyxum* sublineage from Chondrichthyes represent parasites exclusively from cartilaginous hosts and they differ in phylogenetic origin from the relevant generic representatives in bony fishes (Figure 2). 

In contrast to the polychaete- and oligochaete-infecting phylogenetic lineages, members of *Sphaerospora sensu stricto* from sharks cluster together but do not branch basal to all other species in this clade, however, they are positioned basal to representatives infecting amphibians and teleosts (Figure 2). *Parvicapsula* sp. from the bonnethead, *Sphyrna tiburo*, represents the first branch in the subtree harboring all members of the family *Parvicapsulidae*, which were without exception sequenced from teleosts. The other myxozoan genera from Chondrichthyes (*Ceratomyxa* spp. and *Ortholinea* sp.) cluster in two lineages that are characterized by mixed assemblages of parasites from cartilaginous and teleost hosts (Figure 2). These lineages are derived, and within them, species from elasmobranchs do not cluster basal. 

### 3.3. Evolution of Myxozoans in Cartilaginous Fish Hosts: Host-Parasite Co-Diversification and Phylogeography

We used the largest monophyletic dataset of myxozoans from elasmobranchs (23 species of *Chloromyxum*) to better understand the evolutionary history of myxozoans after initial conquest of the new host group and establishment of a two-host life cycle. We investigated whether chloromyxids and their cartilaginous hosts co-diversified and whether geographic barriers impacted on the radiation of *Chloromyxum* spp. in sharks and bathoids. 

Analyses in CoRe-PA determined that parasite and host phylogenies are strongly correlated (Appendix A; Figure 3b). The best scenarios based on fully resolved trees estimated 10 cospeciation events (Appendix A). Partially collapsed parasite trees with polytomies that accounted for the low support of some branches still correlated significantly with host phylogenies, with eight cospeciation events determined (Appendix A) (21 hosts, 23 parasites, in all analyses). All scenarios further indicated that several lineages of *Chloromyxum* invaded elasmobranchs (Figure 3b—basal node), indicating multiple successful establishments. Global fit analyses also supported a cophylogenetic scenario in 14 out of 19 host-parasite pairs, however, with a significant but marginal *p*-value = 0.044, likely due to a limited dataset (19 parasites, 18 hosts; dataset reduced due to missing full-length sequences for several species (excluded)). 

Historical biogeographical reconstructions in S-DIVA showed that *Chloromyxum* spp. in softnose skates (Arhynchobatidae) belonging to the genera *Rioraja*, *Atlantoraja*, *Bathyraja*, and Psammobatis evolved from a common ancestor in a restricted coastal area of South America, i.e., along the coastline of southern Chile, Argentina, Uruguay, and south to central Brazil (zones B and C, Figure 3a). Amongst these, the genera *Bathyraja* and *Psammobatis* speciated most recently with 100% support for a common ancestor in the most southern part of Argentina and Chile (zone C, Figure 3a). Significant agreement regarding phylogenetic and geographic clustering and relevant support for a common ancestor was only detected in one other section of the tree, i.e., in one of the two major lineages of shark hosts, including Squatiniformes (*Squatina*), Squaliformes (*Squalus*, *Centroscymnus*), and Pristiophoriformes (*Pristiophorus*) (Figure 3a). S-DIVA indicates a birth zone for these species and their parasites in the Pacific around Southern Australia (support of 96.5% and 82.3%). However, the present dataset is extremely limited and two of the four species in this group are actually highly migrant, circumglobal species (*Squalus acanthias* and *Centroscymnus coelolepis*). Otherwise, there was no clear correlation between phylogeny and biogeography (Figure 3a).

## 4. Discussion

The nature of the deepest branches in the phylogenetic tree as well as the last common ancestor and host are key questions in myxozoan biology, having wide ramifications for understanding diversity, evolutionary history, and adaptive strategies in this parasite group. Myxozoans are intriguing group of obligate parasites that evolved at the base of metazoans, showing extremely derived genomes [8] and complex life histories (summarized in [3]). Given that it is likely that the vast majority of myxozoan diversity has not yet been sampled and molecular data are only available for approximately 30% of described species [40], our approach represents only a first step for strengthening the base of the tree by adding evolutionary ancient parasite sequence data from some of the first fish hosts available on Earth for secondary host acquisition. While broader sampling is certainly required, we hoped to characterize at least some isolates that pre-date the epoch of within teleost mass-diversification. 

It is likely that ancient chondrichthyan lineages are represented by two subclades, which constitute new host acquisitions in chimaeras (*B. vetusta*) and elasmobranchs (*Chloromyxum* spp.). We hypothesize that both lineages originated and diversified only in cartilaginous hosts, potentially largely prior to the occurrence of teleosts on Earth. This idea is supported by molecular dating of the basal divergence of polychaete- and oligochaete-infecting lineages of myxozoans which was estimated to have occurred 537–447 mya [3] while Chondrichthyes emerged 447–410 mya [4,5], and ray-finned fishes (Actinopterygii) and teleosts 379–340 mya [41]. Furthermore, in contrast to all other chondrichthyan lineages of myxozoans, both *Bipteria* and *Chloromyxum* differ in morphotype and phylogenetic origin from the relevant generic representatives in bony fishes (Figure 1; [14,42,43]), with para- or polyphyletic-lineages being a common problem in the Myxozoa. As an example, *Chloromyxum* spp. known from freshwater teleosts and amphibians are spherical to ovoid with abundant surface ridges while species belonging to the monophyletic group from Chondrichthyes are ovoid to pyriform, with only partial ridges and long posterior appendages in the shape of filaments (reviewed in [43]).

As a surprising result from the present study, myxozoans from elasmobranchs in the *Sphaerospora sensu stricto* clade do not occupy the most basal position in the respective clade, as in the other two annelid-infecting clades. However, they cluster basal to species infecting amphibians and teleosts (Figure 2), indicating that they likely settled in higher vertebrates and teleosts by host switch from Chondrichthyes, thereby reflecting the evolutionary history of these hosts ([3]; see also below). This raises the question if other, evolutionarily even more ancient fish (Agnatha: cyclostomes or the extinct conodonts and ostracoderms) served as initial vertebrate hosts for the first subclade within sphaerosporids. Since this first subclade harbors exclusively marine species [38], hagfish could be a feasible ancestral host group. A single record of a myxozoan in hagfish exists [16], however DNA sequence data is missing. On the other hand, it may well be that this first subclade distinguishes itself from the remainder of sphaerosporids due to an independent historic event of invertebrate host acquisition in an annelid stem group, such as Haplodrilii (basal polychaetes) or Sipuncula (known hosts of myxozoans without DNA sequence records [44]), much earlier in the evolution of sphaerosporids [38], when they first invaded invertebrates. Future investigations into these potential host groups likely bear exciting outcomes. Another surprising finding regarding this particular parasite clade is that, despite screening large numbers of Batoidea, we did not detect sphaerosporids in these hosts, though a single record from rays exists ([45]; no DNA sequence data). 

The phylogenetic position of ceratomyxids and of *Ortholinea* sp. from Chondrichthyes indicates that these chondrichthyan host acquisitions are likely the result of host switches from teleost ancestors. They occur in derived clades with a mixed teleost and chondrichthyan host composition. Furthermore, ceratomyxids in sharks occur in relatively recent shark and ray species (*Carcharhiniformes* and *Myliobatiformes*), which diverged from other groups 193–181 mya and diversified 163–130 mya [22], when teleosts were already roaming the Earth [41]. This shows that, despite bony fishes being predominant in aquatic habitats (at present, teleosts outnumber cartilaginous fish approx. 22 times) cartilaginous fishes can yet be preferred as hosts by some myxozoan species. 

Considering that chondrichthyans represent some of the first vertebrate hosts of myxozoans and data originates from limited host screenings when compared with teleosts, the diversity in this group is remarkable, with 70 taxa known to date. In contrast to elasmobranchs, only two species are known from chimaeras and it is unsure if this is due to a bias in sampling, or if limited diversification of the host group (49 species of Holocephali vs. 1143 species of Elasmobranchii; [22]) could be linked to a lower potential for parasite encounter and diversification [46,47]. The latter would suggest that most of the chondrichthyan host acquisitions happened after the emergence of elasmobranchs. Supportive for this hypothesis is that one of the two species known from chimaeras, *B. vetusta* occupies a unique basal lineage (Figure 2) and morphotype [14] when compared with myxozoans from elasmobranchs, however, more parasite sequence data from chimaeras is required to confirm the proposed independent host entries and evolutionary trajectories. In contrast, the second myxozoan parasite species described from chimaerids is of a morphotype that is common in teleosts which suggests that it was likely acquired by host switch from bony fishes, as stated above. 

The addition of 22 new unique 18S rDNA sequences of myxozoans from cartilaginous hosts allowed for new insights into the origins of myxozoan parasites in sharks, skates, and rays. Most importantly, adding a large number of evolutionarily distinct sequences in basal positions to the phylogenetic tree changed the relationship between the major clades. Of the four major clades known, three are characterized by different invertebrate host groups, Malacosporeans in phylactolaemate bryozoans, polychaete-infecting myxozoans and oligochaete-infecting myxozoans. The life history of members of the fourth clade, *Sphaerospora sensu stricto*, is yet to be confirmed, as a single experimental study exists that identified an oligochaete as a sphaerosporid invertebrate host [48]. The reconciliation of *Sphaerospora sensu stricto* as a sister clade to the polychaete-infecting myxozoans is uncommon but has occasionally occurred previously [49,50] and may suggest that sphaerosporids, like members of polychaete- and oligochaete-infecting clades, emerged as parasites of annelid stem groups [3,38], rather than representing an isolated group parasitizing a different invertebrate group, such as the Malacosporeans. Malacosporeans invade freshwater bryozoans belonging to the Phylactolaemata, and though known to parasitize cyprinid and salmonid fishes, have never been found in Chondrichthyes. It has been hypothesized that the last common ancestor of today’s Phylactolaemata, the radix group of all bryozoans, first evolved in marine environments and only secondarily occupied freshwaters habitats [51]. However, it is uncertain if malacosporeans and their phylactolaemate hosts ever coincided with cartilaginous fishes in marine habitats. To the present knowledge, it is hence unclear if malacosporeans initially parasitized ancient cartilaginous fishes and if extant species lost infections due to the absence of Phylactolaemata in marine habitats. About 5% of all chondrichthyans occur in freshwater habitats, with different evolutionary entries into these environments and some species being completely euryhaline [52]. These may be an interesting target group in the search for ancient malacosporeans in Chondrichthyes.

Our diversity data clearly supports a correlation between high parasite speciation rates and high diversification rates of elasmobranch host groups, which had recently been observed in teleosts [3]. Our analyses spotlight the *Carcharhiniformes* (ground sharks) as a species-rich shark family, from which 32 out of the total of 70 parasite taxa have been reported. Myxozoan-infected ground sharks are to a large extent migratory species. Migration is commonly assumed to enhance the geographical spread of parasites and can expose animals to a higher diversity of infective stages as they move between breeding and wintering or feeding grounds [53,54]. Resident bird species, for example harbor lower parasite richness of nematodes and helminths in general compared to migratory species [55,56]. Apart from being one of the most diverse families of elasmobranchs, migratory behavior may serve as an additional reason for ground sharks being attractive hosts for myxozoans. Higher numbers of myxozoans were present in highly migratory elasmobranchs such as groud sharks, though additional data is required to proof significant correlations.

Historical biogeographical reconstructions of the evolutionary history of myxzoans and their elasmobranch hosts in the monophyletic group of chloromyxids identified the Atlantic as the birthplace of softnose skates (*Arhynchobatidae*), skates (*Rajidae*), and their parasites. This is however based on a dataset of species (genera *Rioraja*, *Atlantoraja*, *Bathyraja*, *Psammobatis*, *Dipturus*, *Rostroraja* and *Beringraja*) inhabiting the coasts of the Atlantic (zones A–E in Figure 3a) and clustering together in phylogenetic analyses. In remains unclear whether their phylogeny mirrors the speciation pattern of their hosts or in fact the biogeographic history of settlement of these species in the Atlantic. Rifting in the central Atlantic occurred 220–198 mya, with seafloor spreading beginning ca. 200 mya in Georgia/North and South Carolina and 180 mya in Massachusetts/Nova Scotia, while the South Atlantic began to open 120 mya [57,58]. These dates are in agreement with the emergence of the chondrichthyan hosts relevant to this analyses (Figure 3a; [22]) and likely that of their chloromyxids (Figure 3b). However, it would be of particular interest to enrich the *Chloromyxum* spp. dataset with closely related ray species from outside the Atlantic to elucidate these relationships in more detail and challenge the present biogeographic observations. The birth zone of Squatiniformes (*Squatina*), Squaliformes (*Squalus*, *Centroscymnus*), and Pristiophoriformes (*Pristiophorus*) and their parasites, in the Pacific around Southern Australia, estimated by S-DIVA, is likely artificial as two of the four species are highly migrant and we believe that additional species from these shark orders are required to confirm or (likely) refute this pattern. The general lack of correlation between phylogeny and biogeography in older lineages and the estimated panglobal ancestor of chloromyxids in chondrichthyans may well be explained by the fact that many chondrichthyan hosts and their myxozoans emerged in Panthalassa, before the present tectonic arrangement of landmasses [58].

The major finding of this study is that a strong correlation exists between host and parasite phylogenies in chloromyxids from elasmobranchs, even after collapsing the branches of the parasite tree that showed low bootstrap support, and by support from non-tree-based methods. All co-phylogenetic scenarios further suggested multiple establishments of different *Chloromyxum* spp. in the new host group. Repeated transmissions would require accessibility or frequent physical contact of parasites and elasmobranchs. The two ancient lineages of chondrichthyan myxozoans (*B. vetusta* and *Chloromyxum* spp.) form spores exclusively in the gall bladders of their hosts, indicating a potential route of entry and mechanism of new host acquisition. Chimaeroids and ancient sharks are predatory, eating primarily hard foods that they crush with their tooth plates. Their diet consists primarily of benthic invertebrates including bivalves, gastropods, various crustaceans, polychaetes, and echinoderms [59,60]. It appears that trophic transmission followed by migration of the parasites to the gall bladder via the common bile duct represents a likely pathway of acquisition of the new host. The bile is considered an immunologically privileged site where the hosts’ immune system has reduced surveillance capability [61] and hence represents a close-by protective niche on ingestion, where the parasite was able to survive despite the advanced immune system capabilities of vertebrates which can mount parasite-specific adaptive responses [62]. Thereby survival in a new host group is considered favorable [63,64] and compensates the initial costs related to generalism [65]. The feed-integration hypothesis receives further support from myxozoan infection trials in teleosts. Gills and skin were identified as common portals of myxozoan entry into fish, however, the buccal cavity and the digestive tract are other evidenced invasion sites [66]. The myxozoan *Thelohanellus hovorkai* is not only able to use digestive epithelia for entry into common carp, but even develops higher infection levels after intubation than following bath exposure [67], despite its target organ being the skin. 

On acquisition of the chondrichthyan host, myxozoans successfully colonized a host of a higher trophic level. Johnson et al. (2010) [68] aptly termed the incorporation of predators as hosts into parasite life cycles ‘ghosts of predators past’. In such an ‘upward incorporation’ [65], parasite life cycles lengthen by adding a new host higher up the food chain, and usually this host subsequently becomes the definitive host, while the original definitive host becomes an intermediate host, with a prolonged larval stage [69]. This succession is explained by host size, as larger hosts can accommodate larger adult parasites and produce more offspring. Hence, a priori it might be expected that complex life cycles should be observed in derived taxa. Blaxter (2003) [70], however, reviewed evidence from different parasite groups such as gnathostomes or ascarid nematodes that convincingly demonstrates that there appears to be no greater barriers to moving from vertebrate to nonvertebrate hosts than vice versa. This pattern of host usage suggests that their recruitment is based on what is adaptive to the parasite, and not restricted by phylogenetic history [70]. Myxozoans likely maintained invertebrates as their definitive host as they remain microscopic parasites throughout their whole life cycle, even though they sometimes produce large plasmodia in fish (e.g., *Kudoa thyrsites*, *Thelohanellus kitauei*) or in invertebrates (e.g., *Buddenbrockia plumatellae*), which harbor millions of individual spores. However, myxozoans never show cell differentiation, organ formation or development into larger individuals, and sexually reproducing stages still appear to be restricted to the original invertebrate hosts {reviewed by [71]). This likely allowed them to switch between completely different host groups several times, acquiring e.g., platyhelminth [10,11,13] or mollusk [12] invertebrate hosts by switches from secondary fish hosts [3]. It is likely that chondrichthyan or teleost host acquisitions were not exploited with regard to an advantage for parasite growth in large vertebrates compared with smaller invertebrate hosts, but rather because they facilitated alternative transmission and dispersion strategies and provided diverse new niches in different host organs, leading to massive diversification of myxozoans in these hosts and their organs [3,24,39].

## 5. Conclusions

In order to reconstruct the evolution of life cycle complexity in the Myxozoa and study the patterns of diversification in the newly acquired vertebrate host, we more than doubled existing 18S rDNA sequence data of myxozoan parasites from evolutionarily ancient fishes belonging to the Chondrichthyes. Our results demonstrate that the oldest lineages of oligochaete- and polychaete-infecting myxozoans evolved in cartilaginous fishes. Furthermore, we provide the first evidence that they speciated as a function of host diversification, as well as biogeographic changes, with phylogeny clearly mirroring the formation and settlement of host species in the Atlantic Ocean. We propose feed-integration of infected invertebrates as a likely mechanism for the establishment of life cycle complexity, and migration and initial development in the gall bladder/bile of chondrichthyans as an immunologically privileged site, confirmed in two independent ancient lineages. The addition of new sequences in basal positions of the myxozoan tree resulted in a change in the organization of the major myxozoan phylogenetic lineages and shows that data from ancient vertebrates embodying large periods of unique evolutionary history are extremely useful for investigating the initial settlement and diversification of myxozoans in vertebrates. Thereby, agnaths offer promising future perspectives as they represent the oldest vertebrate group and are evolutionarily even more distinct than cartilaginous fishes, while they are almost unexplored to date as hosts for this enigmatic group of cnidarian parasites.

## Figures and Tables

**Figure 1 biology-09-00010-f001:**
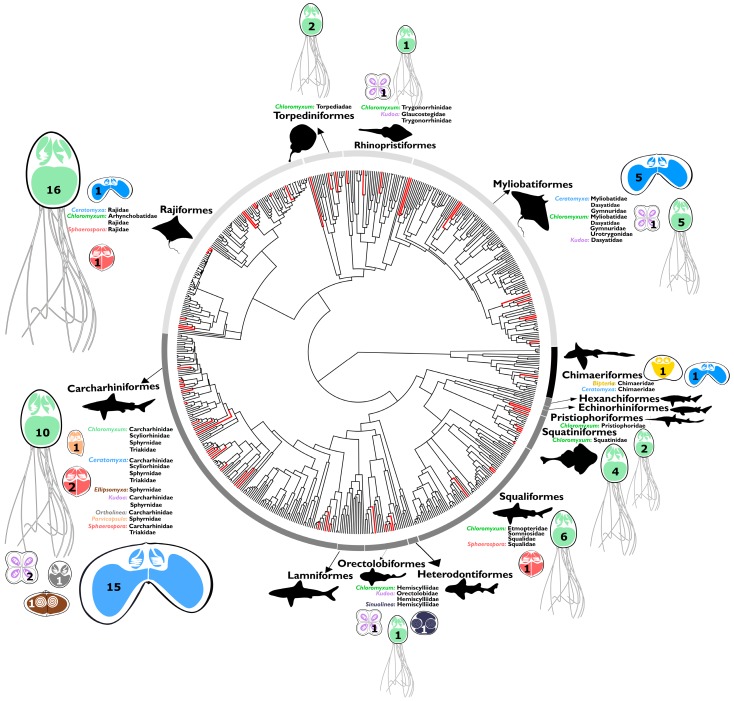
Representative taxon-complete tree of Chondrichthyes based on a DNA supermatrix of 15 coding and non-coding DNA regions (60,121 bp; [22]), indicating fish species with myxozoan infection (red lineages) and distribution of myxozoan genera in different families of chimaeras, sharks, rays, and skates. Number within myxozoan spores and their size is indicative of parasite species found in the relevant lineage.

**Figure 2 biology-09-00010-f002:**
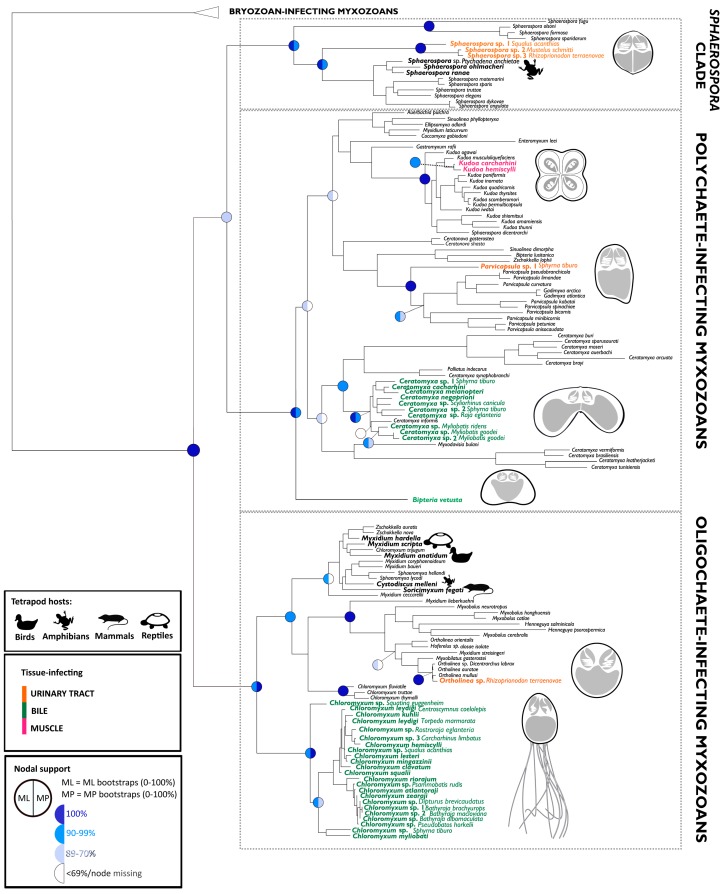
Phylogenetic tree of myxozoans based on 18S rDNA sequences, indicating the four main lineages (Malacosporea = bryozoan-infecting myxozoans, *Sphaerospora sensu stricto*, polychaete-infecting myxozoans, and oligochaete-infecting myxozoans) and the origin of myxozoan lineages in sharks, rays and skates (seven independent lineages, colored species names). Note *Bipteria vetusta* and *Chloromyxum* spp. represent the most basal lineages in their respective clades. These inhabit the bile and, to the present knowledge, occur only in Chondrichthyes.

**Figure 3 biology-09-00010-f003:**
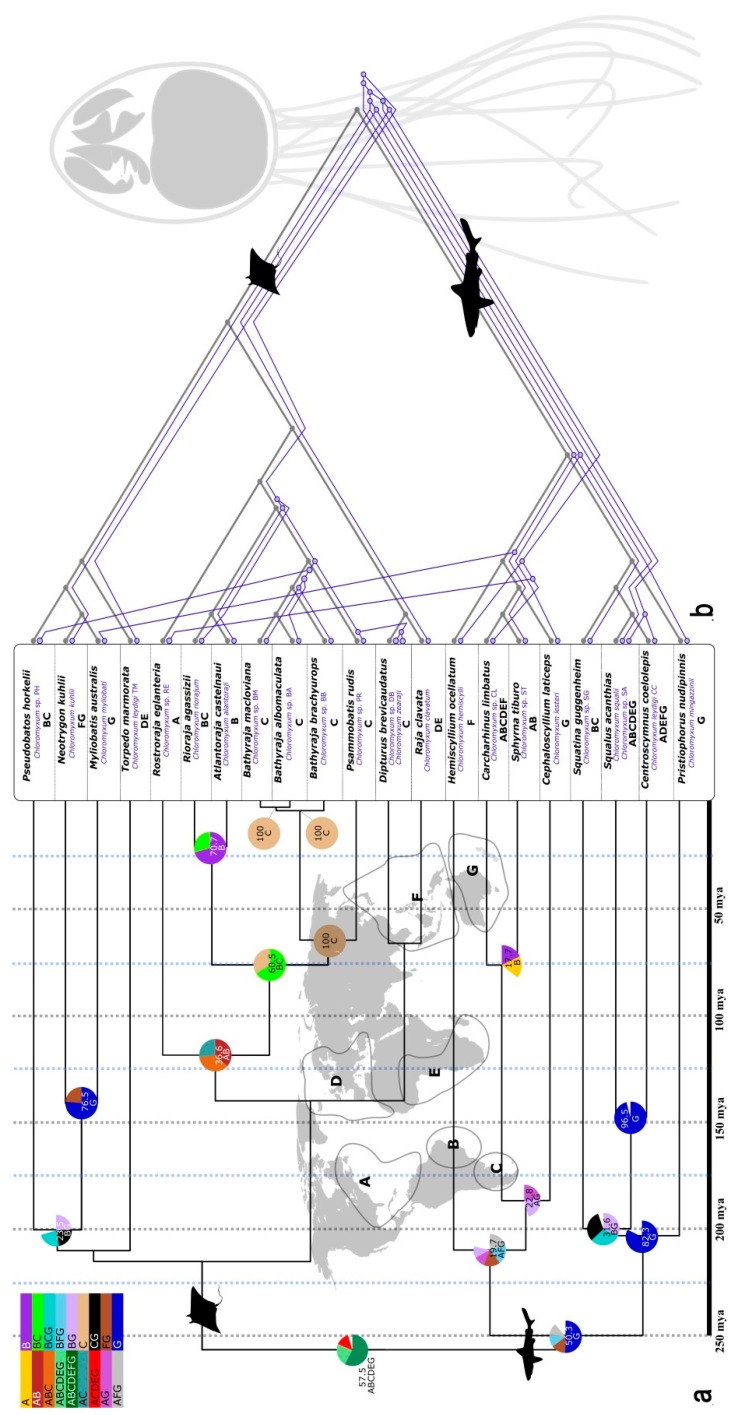
Phylogenetic trees of *Chloromyxum* spp. and their respective cartilaginous fish hosts: (**a**) timed tree of chondrichthyans with dated species emergence, based on multi-gene phylogenetic analysis and molecular clock analyses [22], indicating ancestral states of geographic–phylogenetic character correlations determined by dispersal-vicariance analyses (S-DIVA). Geographic areas defined by letters A-G in coastal zones around the world (background of Figure 3a), numbers of nodes show highest support (%) for an origin in a certain area. (**b**) Result of co-phylogeny analyses of chondrichthyans (black cladogram; same tree as in Figure 3a) and their *Chloromyxum* spp. (blue cladogram, mapped to host; based on 18S rDNA data), using CoRe-PA, showing significant overlap of host and parasite phylogenies, with only three estimated host switches.

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
