# Peer review of "Mechanisms and Drivers for the Establishment of Life Cycle Complexity in Myxozoan Parasites"

_biology, 2020, doi:10.3390/biology9010010_

Round 1
Reviewer 1 Report
The authors present new genetic data they leverage to construct a phylogeny of myxozoan parasites, with the aim of understanding host switches and the evolution of complex life cycles. Their study contributes considerably to understand myxozoan phylogeny, and highlights potential avenues of host switching/transition to complex life cycles, including via the gall bladder of predatory species. I found the paper to be generally well-written and compelling, but the methods seemed very spare. Fleshing out the justification of some of the key choices in the analysis (especially the cophylogeny comparison) would make this article a more useful contribution.
The authors use broad migration categories to assess a correlation between parasite species richness and migration range of host species. They find no significant pattern, and I would like some additional text regarding the decision to use migration categories rather than, for example, maximum distance traveled during migration. If they feel they would lose statistical power by taking that approach, because they would have to omit large numbers of studies, that should be made clear in the main text. It would also be useful for the authors to highlight what data would be most useful here—are the common ways of assessing migration patterns useful to this sort of analysis?
I appreciate that the authors included supplementary tables of the key information, but it would be more helpful if these were appended as tables that could be easily read by software so that the analysis could be repeated easily with more data/different categorization decisions. Including the code used to perform the analysis would also be valuable.
The details of the statistical analyses are quite sparse in the main text. For example, there are many models for comparing host and parasite phylogenies, and some discussion of why CoRe-PA was chosen, and whether similar results would have been obtained with different software, would seem to be warranted (see methods in Sweet et al. 2016, Biological Journal of the Linnean Society or Anthony et al. 2017 Virus Evolution).
Do the authors have any ideas regarding why they didn’t find any sphaerosporids in Batoidea, despite the previous record?
Minor points:
Line 165 and 289-290: There seems to be verbs missing from this sentence.
Author Response
We would like to thank the referee for the valuable criticism and respond to the questions raised in the following section:
The authors present new genetic data they leverage to construct a phylogeny of myxozoan parasites, with the aim of understanding host switches and the evolution of complex life cycles. Their study contributes considerably to understand myxozoan phylogeny, and highlights potential avenues of host switching/transition to complex life cycles, including via the gall bladder of predatory species. I found the paper to be generally well-written and compelling, but the methods seemed very spare. Fleshing out the justification of some of the key choices in the analysis (especially the cophylogeny comparison) would make this article a more useful contribution.
Authors response: We added additional text explaining our choices regarding the methods used for cophylogeny and migration analyses. We further added PARAFIT as a topology-based approach (versus the performed event-based analyses in CoRe-PA) to the cophylogeny analyses, to eliminate doubts about the support for codiversification of parasites and hosts, from other methods.
The authors use broad migration categories to assess a correlation between parasite species richness and migration range of host species. They find no significant pattern, and I would like some additional text regarding the decision to use migration categories rather than, for example, maximum distance traveled during migration. If they feel they would lose statistical power by taking that approach, because they would have to omit large numbers of studies, that should be made clear in the main text. It would also be useful for the authors to highlight what data would be most useful here—are the common ways of assessing migration patterns useful to this sort of analysis?
Authors response: Additional text on the choice of migration categories was provided together with a comparison with commonly used approaches. We tried a number of analyses and categories here and some of them even provided significant outcomes but we feel this would be modelling of the data towards a significant result. We know from transponder studies that some sharks of the same species differ considerably in the distance they travel during seasonal migrations. Since we don’t know the exact trajectories of our particular hosts, we defined ranges. Putting species into commonly used categories of 'migratory, nomadic or resident' left us with a too small dataset for analysis in the nomadic category, and hence to be able to compare all three categories. Furthermore, the distance travelled by some species with seasonal migration is often larger per year than that travelled by nomadic species, so they could principally encounter more species of parasites along their routes. We thought that distance categories may be the easiest to handle and designed them in a way that enough data was available for each category. These explanations were provided to the reader in a shortened version.
I appreciate that the authors included supplementary tables of the key information, but it would be more helpful if these were appended as tables that could be easily read by software so that the analysis could be repeated easily with more data/different categorization decisions. Including the code used to perform the analysis would also be valuable.
Author’s response: We provided the data in the MS word table as an MS excel file. This can be easily imported into most statistical packages.
The details of the statistical analyses are quite sparse in the main text. For example, there are many models for comparing host and parasite phylogenies, and some discussion of why CoRe-PA was chosen, and whether similar results would have been obtained with different software, would seem to be warranted (see methods in Sweet et al. 2016, Biological Journal of the Linnean Society or Anthony et al. 2017 Virus Evolution).
As stated above, we added PARAFIT as a topology-based approach (versus the performed event-based analyses in CoRe-PA) to the cophylogeny analyses, to eliminate doubts about the support for codiversification of parasites and hosts, based on other methods.
Do the authors have any ideas regarding why they didn’t find any sphaerosporids in Batoidea, despite the previous record?
Author’s comment: Unfortunately not. The previous report provides only a morphological description, however, this species may still have a different phylogenetic origin than the Sphaerospora spp. from sharks which cluster in the sphaerosporid clade, such as e.g. Sphaerospora testicularis from seabass (clusters in the polychaete-infecting clade). We are trying to obtain batoid hosts from the region where this Sphaerospora was reported.
Minor points:
Line 165 and 289-290: There seems to be verbs missing from this sentence.
Author’s response: Missing verbs were added.
Reviewer 2 Report
This is truly an excellent paper on a very well executed study of great significance to scientists who study parasitism, invertebrate biology, evolution, and especially cnidarian/myxosporan diversity. With only a very few minor errors, the text is superbly written both in technical form as well as clear readability and organizational integrity. I have marked a couple of very minor editing suggestions on the attached reviewer-marked copy.
The authors have done a commendable job of covering the pertinent literature. This paper, in addition to providing new data and analysis, will serve as a very valuable literature review of this fascinating and important group of parasitic cnidarians.
I do have a couple of suggestions regarding the Materials and Methods:
1) Would it be possible to add the geographic coordinates of collections sites? The Gulf of Mexico near Sarasota and the Charleston Harbor area of the Atlantic coast are rather broad and somewhat diverse both biologically and in terms of hydrology and sedimentology. Also, having access to this information may help others later who want to collect in these areas in search of particular species. This is just a suggestion for the authors, who may have various reasons for not including this information.
2) In the Materials and Methods, there are some statements, apparently intended to inform the reader about the authors' reasons for including certain subjects, which are not strictly results of this study, but might be more appropriately discussed in the Discussion section. Again, this is just a suggestion, but I ask the authors to consider it.
In summary, this is a superbly presented paper on a very important and novel study, with valuable literature review and analysis. I recommend that it be published quickly following very minor revision.

Author Response
We would like to thank the reviewer for the positive feedback and respond to the two queries raised:
1) Would it be possible to add the geographic coordinates of collections sites? The Gulf of Mexico near Sarasota and the Charleston Harbor area of the Atlantic coast are rather broad and somewhat diverse both biologically and in terms of hydrology and sedimentology. Also, having access to this information may help others later who want to collect in these areas in search of particular species. This is just a suggestion for the authors, who may have various reasons for not including this information.
Author’s response: The hosts were obtained on fishing trips of commercial boats (Charleston Harbour) or research vessels (Sarasota) which covered fairly large areas by netting various times, in adjacent fishing sections. We are hence unable to provide an exact sampling site by coordinates.
2) In the Materials and Methods, there are some statements, apparently intended to inform the reader about the authors' reasons for including certain subjects, which are not strictly results of this study, but might be more appropriately discussed in the Discussion section. Again, this is just a suggestion, but I ask the authors to consider it.
Author’s response: We integrated into this section only references that provide support for the choice of methods used. We feel this is necessary here and referee 1 required further explanations including additional references to be added. We hence kept these in the MM section.
Reviewer 3 Report
I'm not often recommending papers to be published "as is", but I think this warrants an exception barring minor spell checks.
Author Response
Author's response: Thank you. We have gone through an additional round of spell checking and hope this is now okay.